# Can Circulating Tumor DNA Support a Successful Screening Test for Early Cancer Detection? The Grail Paradigm

**DOI:** 10.3390/diagnostics11122171

**Published:** 2021-11-23

**Authors:** Oscar D. Pons-Belda, Amaia Fernandez-Uriarte, Eleftherios P. Diamandis

**Affiliations:** 1Department of Pathology and Laboratory Medicine, Mount Sinai Hospital, Toronto, ON M5G 1X5, Canada; oscardavid.pons@sespa.es (O.D.P.-B.); afernandez@catlab.cat (A.F.-U.); 2Department of Laboratory Medicine and Pathobiology, University of Toronto, Toronto, ON M5S 1A8, Canada; 3Department of Clinical Biochemistry, University Health Network, Toronto, ON M5G 2N2, Canada

**Keywords:** circulating tumor DNA, liquid biopsy, cancer screening, early cancer detection, molecular analysis, clonal hematopoiesis, positive predictive value

## Abstract

Circulating tumor DNA (ctDNA) is a new pan-cancer tumor marker with important applications for patient prognosis, monitoring progression, and assessing the success of the therapeutic response. Another important goal is an early cancer diagnosis. There is currently a debate if ctDNA can be used for early cancer detection due to the small tumor burden and low mutant allele fraction (MAF). We compare our previous calculations on the size of detectable cancers by ctDNA analysis with the latest experimental data from Grail’s clinical trial. Current ctDNA-based diagnostic methods could predictably detect tumors of sizes greater than 10–15 mm in diameter. When tumors are of this size or smaller, their MAF is about 0.01% (one tumor DNA molecule admixed with 10,000 normal DNA molecules). The use of 10 mL of blood (4 mL of plasma) will likely contain less than a complete cancer genome, thus rendering the diagnosis of cancer impossible. Grail’s new data confirm the low sensitivity for early cancer detection (<30% for Stage I–II tumors, <20% for Stage I tumors), but specificity was high at 99.5%. According to these latest data, the sensitivity of the Grail test is less than 20% in Stage I disease, casting doubt if this test could become a viable pan-cancer clinical screening tool.

## 1. Introduction

Circulating fetal DNA was originally discovered by clinical chemist Dr. Dennis Lo in the 1990’s (Figure 1), who found that in the serum/plasma of pregnant women there is 5–10% of circulating free DNA (cfDNA) of fetal origin [1]. This discovery led to the application of cfDNA for the non-invasive detection of fetal abnormalities by using maternal blood (non-invasive prenatal diagnosis) [2]. Later, Dennis Lo and others identified cell-free DNA in the serum of patients with cancer. Some of this DNA is of tumor origin, termed circulating tumor DNA (ctDNA). Since then, it has been hypothesized that it may be possible to use what is now widely known as a “liquid biopsy” (essentially blood taking) for the diagnosis and prognosis of cancer by analyzing ctDNA. ctDNA has now emerged as the latest and a highly promising new cancer biomarker for diverse clinical applications [3,4,5].

A liquid biopsy involves obtaining blood or other fluids and then analyzing the extracted DNA, usually by diverse molecular techniques. Liquid biopsy is minimally invasive, and it can be applied to analyze ctDNA, tumor cells, exosomes, and other components. The preferred sample is blood/plasma, but other fluids such as cerebral spinal fluid, saliva, synovial fluid, ascites fluid, stool, and urine can also be used [6].

ctDNA is fragmented DNA that originates from dying cancer cells and has been shed into the bloodstream [7]. Importantly, it is well-established that the amount of ctDNA is proportional to tumor size (tumor burden) and is related to other clinicopathological parameters such as stage, lymph node infiltration, local and distant metastasis, and disease-free and overall survival [8,9]. Thus, ctDNA carries strong prognostic information [10,11,12,13,14,15,16].

Healthy cells also release what is known as cfDNA into the bloodstream. The amount of cfDNA that can be extracted from the blood of normal people is about 1–10 ng/mL.

## 2. Liquid Biopsy

Usually, a tube of blood is obtained from a patient, and DNA is extracted from plasma. This DNA can then be used for analysis with various techniques, including whole-genome sequencing, whole-exome sequencing, targeted sequencing of cancer-associated genes, or by looking for gene fusions, copy number variations, and DNA methylation status [6]. While performing liquid biopsies, it is important to consider some caveats. One caveat is that blood contains both normal DNA originating from diverse normal tissues (cfDNA) and genetically altered DNA that is expected (as per the current wisdom) to originate from cancer cells (ctDNA). However, this is not always the case [17]. Genetic analysis of DNA must be able to discriminate between the normal DNA and the cancer DNA by looking for mutations, epigenetic, and other molecular changes. The ratio of cancer to normal DNA, expressed as a percentage, is known as “mutant allele fraction” (MAF). For example, a MAF of 0.1% means that for every 1000 DNA molecules in the circulation, one is contributed by cancer cells, and 999 are contributed by normal cells. It is known that the higher the tumor volume, the higher the mutant allele fraction [6] and the easier the analysis of the extracted DNA. It is evident that when the MAF is very low, such as <0.1% or <0.01%, there is a need for special and highly sensitive techniques to analyze the ctDNA, which is a minute fraction in comparison to normal DNA (cfDNA) [6].

## 3. Clinical Applications of ctDNA

ctDNA is an excellent prognostic marker for cancer. For example, in a 2016 study of 230 stage 2 colon cancer patients, 100% of those who had detectable ctDNA right after the surgery relapsed. However, more than 90% of those who were negative for ctDNA did not relapse [13]. Since then, the prognostic value of ctDNA was repeatedly confirmed [8,9,10,11,12,13,14,15,16,18,19]. Similar to other tumor markers, ctDNA can also be used to monitor disease progression. One example is breast cancer patients who have been treated at various intervals with chemotherapy [20]. ctDNA was superior to CA15.3 (the classical breast cancer biomarker) and circulating tumor cells in tracking disease progression and regression.

An important application of circulating tumor DNA is monitoring the success of treatment [21]. The patient is treated initially with certain therapy while he/she is monitored with liquid biopsy for the presence and intensity of mutations in ctDNA. The ctDNA analysis may show that the mutations have been reduced in numbers and/or intensity, indicating that the therapy is working and should continue to be administered. On the other hand, for other patients, it may be shown that there is progression because the intensity and the number of mutations are increased, thus alerting the clinician to select a new therapy [21]. The amount of ctDNA, as mentioned earlier, correlates quite well with the size of the tumor. Larger amounts of ctDNA correlate with larger tumors, advanced tumors, or later-stage tumors.

## 4. Early Cancer Diagnosis

While the utility of ctDNA for monitoring the success of treatment and prognosis is unquestionable, and the marker is now being used in the clinic, there is a debate in the literature if ctDNA can be used for the most impactful application of early cancer detection [22]. A company called Grail invested billions of dollars in developing methods for early cancer detection by using ctDNA. For more details on Grail, see reference [23,24,25]. In short, Grail has embarked on a large project aiming to enroll 10,000 individuals with cancer and 3000 healthy individuals and analyze their cfDNA, to find differences between cancer and non-cancer patients, thus developing an early cancer detection test, aided in decision-making by artificial intelligence. The test has detection capability for any cancer (a pan-cancer test) and is good enough to identify the cancer primary lesion with about 80% accuracy [25]. The techniques that Grail uses to analyze ctDNA include targeted sequencing of about 500 genes previously associated with cancer, whole-genome sequencing, and copy number variations, whole-genome bisulfite sequencing to study methylation status [22], while other investigators are using DNA fragmentation to study the sizes of ctDNA [26,27,28,29]. Academic investigators have also published on the detection of early cancer by using ctDNA [13,30]. In one of these papers, the authors claimed sensitivities for detecting cancer of around 50–60% at 95% specificity, which are good numbers if they can be independently reproduced. Unfortunately, these and all other published studies suffer from the limitation that these investigators used clinically detected cancers with relatively high mutant allele fractions of 0.1% to 1%, thus predictably over-estimating sensitivity [31]. The approach used by the Nickolas Papadopoulos group (Figure 2) included combination of ctDNA with classical circulating biomarkers for early detection of cancer [30]. These authors claimed sensitivity between 70–90% at 98% specificity, which are good numbers if they can be reproduced [30]. A company called Thrive attracted $110 million to commercialize this approach.

## 5. Empirical Calculations Challenging the Clinical Utility of Grail and Similar Tests for Early Cancer Diagnosis

Here, we will provide a summary of our detailed and previously published calculations, mostly based on published empirical data, which show that ctDNA tests for the early diagnosis will likely have limited sensitivity for small, asymptomatic tumors. For more details, see our previous analyses [22,32,33,34].

The amount of cfDNA in the plasma of normal individuals falls within the range 1–10 ng/mL (average 5 ng/mL) [34]. Assuming a molecular mass of DNA of approximately 2 × 10^12^ Da, 5 ng/mL of DNA equates to approximately 1500 whole human genomes per mL (6000 genomes per 4 mL of plasma). According to these data, when the mutant allele fraction of cancer DNA drops below 0.01% (one cancer genome admixed with 10,000 normal genomes), then the use of 10 mL of blood (4 mL of plasma) will likely not contain a single cancer genome for molecular analysis, thus rendering the diagnosis of cancer impossible, due to sampling error.

In patients with small tumors, we also used other reported tumor measures to calculate the approximate amount of cancer or normal DNA in the circulation [35,36]. Table 1 summarizes our calculations (these are either experimental data or the numbers were calculated by extrapolation), assuming proportionality between tumor volume and percent fraction of mutant DNA, as suggested by Abbosh et al. [36]. It is reported in the literature that a tumor of approximately 1 cm^3^ in volume has a wet weight of 1 g, contains 10^9^ cells [37], and has an approximate diameter of 1.2 cm (assuming a spherical nodule). The table demonstrates that when the mutant allele fraction drops below 0.01% (one tumor DNA molecule admixed with 10,000 normal DNA molecules), then 10 mL of blood (4 mL of plasma) will likely contain less than one cancer genome, rendering diagnosis impossible. We reached the same conclusion using other independent cancer data (not shown) and by modeling pregnancy at various gestational ages, assuming that the fetus resembles a tumor [32,33,34].

Dr. Steven Narod reported the likelihood of progression of small breast tumors and correlated the findings with the known sensitivity of mammographic screening [38,39] (Table 1). If we set a clinical screening requirement to detect cancers that are >6% likely to progress and are currently missed by mammography, then a 5 mm diameter tumor would be a realistic and clinically relevant early detection goal. As we have determined through our calculations (Table 1), this goal is not likely to be achieved by the Grail test and similar other technologies. More ambitious goals, such as the detection of 1 mm diameter tumors, need to be balanced with unfavorable consequences, such as over-diagnosis and over-treatment [40]. A summary of the merits and disadvantages of cancer screening programs is shown in Table 2 [34].

The presented empirical data suggests that current ctDNA-based diagnostic methods could predictably detect tumors of sizes greater than 10–15 mm in diameter. Tumors of such sizes are currently detectable through imaging [39]. Our calculations also pinpoint a MAF of 0.01% as the detection limit of current Grail-like methods. A summary of our conclusions is shown in Figure 3.

## 6. Additional challenges with Grail and Related Technologies

In screening programs, the sensitivity of a cancer test is not the only important characteristic; specificity could be as important or, in certain screening scenarios for rare tumors, even more important than sensitivity. It is now well-established that mutations and other genomic alterations can be found in normal tissues [17] and in circulating DNA from normal people, especially in the precancerous conditions called clonal hematopoiesis [41,42,43,44]. Discussion of other caveats with screening, including prevalence, overdiagnosis, overtreatment, and tumor dynamics can be found in the associated references [40,45].

## 7. Experimental Data from Grail That Support our Predictions

Recently, Klein et al. from the Grail group presented a large validation study of a targeted methylation-based multi-cancer early detection test using an independent patient validation set [25]. This study was the third and final part of the series of studies by the same authors/company that incorporates their best analytics thus far, which is methylation sequencing, in combination with artificial intelligence. This was a case-control study, not a simulated screening study. The authors report a specificity of 99.5%, which is impressive if it holds true in screening settings. Sensitivity was around 52% overall but was strongly dependent on the stage, as expected from the previous discussion. Sensitivity was 17% for Stage I, 40% for Stage II, 28% for Stages I-II, and 84% for Stages III-VI disease. These data prompted the authors to conclude that their results support the feasibility of the blood-based multi-cancer early detection test as a complement to existing single-cancer screening tests. In fact, the British Ministry of Health contracted Grail to screen 200,000 Britons for early cancer detection (these data are pending).

In our view, the claim that such tests are close to reaching the clinic is premature, mainly because the test’s sensitivity is poor for early-stage tumors. Based on the way patients were enrolled (clinically symptomatic disease; a case-control study), we can safely predict that about 9 out of 10 small, asymptomatic tumors, which are amenable to curative therapies, will likely be missed (a more realistic sensitivity will likely be around 10%, as described earlier [46] by the same authors).

The authors extrapolated what will be the positive predictive value (PPV) of such tests (PPV = the chances that the disease is present if the test is positive). If the sensitivity is assumed to be 10% for early cancers under a screening scenario, at 99.5% specificity, and a hypothetical prevalence of cancer in the general screening population of either 1% (for fairly common cancers) or 0.1% (for fairly rare cancers), the PPV will be 17% in the first case and 1.7 % in the second case. We seriously question whether a successful screening program for cancer can be sustained with such low PPVs. The false positivity rates will lead to a rather large number of non-cancer patients who will undergo additional, unnecessary, and probably harmful testing [40].

## 8. Conclusions

ctDNA is a new and exciting cancer biomarker with many clinical applications. Based on our previous analyses, this test could detect some small, asymptomatic tumors, which are amenable to cure. We believe that the vast majority of small but clinically significant tumors will be missed under a screening scenario. Based on our calculations, we set the tumor size for detectability above 10–12 mm in diameter (Figure 3).

Grail’s current and previous clinical trial data, including those reported recently [25], do not warrant population screening based on analysis of methylation patterns, even if such tests are highly specific. The problem lies in the low sensitivity for detecting early, asymptomatic tumors. We further believe that physicians will be reluctant to operate on patients who do not have imaging-confirmed masses. We are not very optimistic about new advances in this area since the problem is not so much the limitations in techniques but the availability of ctDNA in sufficient amounts with a simple blood draw. Much larger blood draws may partially provide a solution, but these are unlikely to be acceptable to the testing population. The data to be generated through the Grail/UK Government collaboration will tell us if Grail has finally found, or missed, the trail that leads to early cancer detection.

## Figures and Tables

**Figure 1 diagnostics-11-02171-f001:**
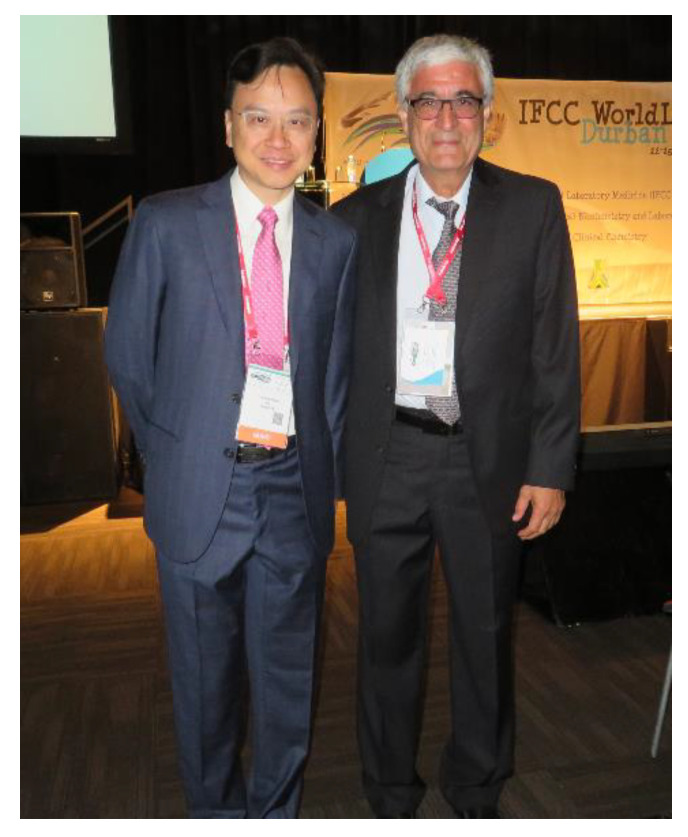
Dennis Lo with one of the authors at the 2017 IFCC Conference in Durban, South Africa.

**Figure 2 diagnostics-11-02171-f002:**
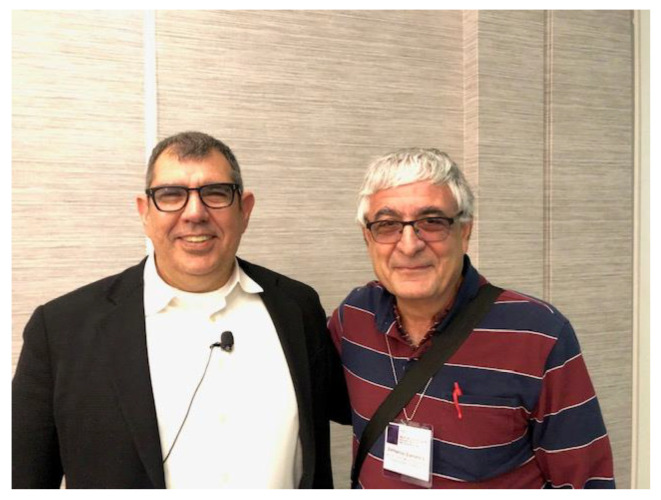
Nickolas Papadopoulos with one of the authors at the 2019 AACR Conference in Hawaii.

**Figure 3 diagnostics-11-02171-f003:**
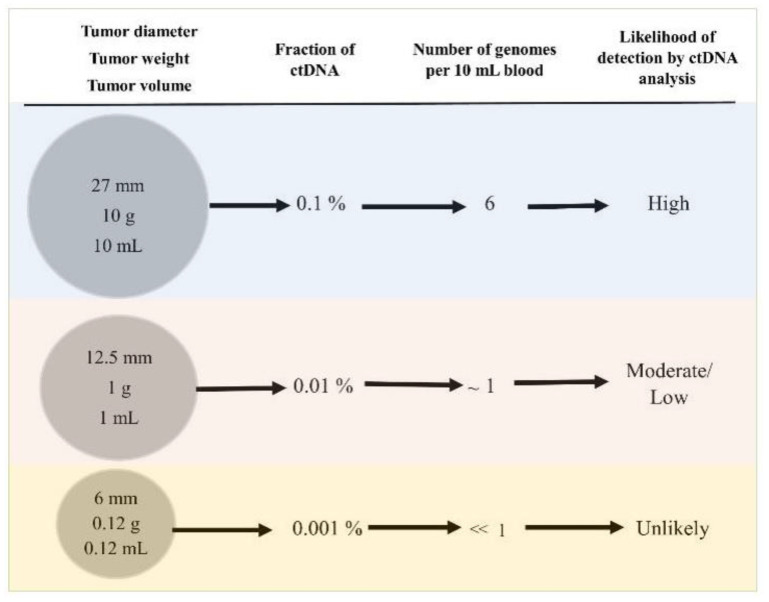
Tumor characteristics and related circulating tumor DNA (ctDNA) parameters. For discussion see text.

**Table 1 diagnostics-11-02171-t001:** Tumor characteristics reported in the literature or calculated by extrapolation.

Tumor Diameter, mm	Tumor Weight, mg	Tumor Volumen mL (cm^3^)	Number of Cancer Cells	Percentage Fraction of Mutant ctDNA	Number of Cancer Genomes per 10 mL of Blood	Chance of Progression ^c^	Mammographic Screen Sensitivity ^d^
27	10,000	10 ^a^	10,000,000,000	1:1000	6	-	-
12.5	1000	1 ^b^	1,000,000,000	1:10,000	0.6	-	-
10	500	0.5	500,000,000	1:20,000	0.3	50%	91%
8	250	0.25	250,000,000	1:40,000	0.15	25%	-
6	125	0.12	125,000,000	1:80,000	<0.1	-	-
5	62	0.06	62,000,000	1:160,000	<0.1	6%	26%
4	31	0.03	32,000,000	1:320,000	<0.1	-	-
3	16	0.015	16,000,000	1:640,000	<0.1	-	-
2.4	8	0.007	8,000,000	1:1,300,000	<0.1	-	-
2	4	0.0035	4,000,000	1:2,600,000	<0.1	-	-
1.5	2	0.0017	2,000,000	1:5,200,000	<0.1	-	-
1.1	1	0.0008	1,000,000	1:10,000,000	<0.1	0.05%	-

ctDNA: circulating tumor DNA. ^a^ As reported by Abbosh et al. [36]. ^b^ As reported by Del Monte [37]. ^c^ As reported by Narod and others [38,39]. ^d^ As reported by Wedon-Fekjaer et al. [39]. Adapted from ref. [34].

**Table 2 diagnostics-11-02171-t002:** Possible Benefits and Harms of Population Screening.

Benefits	Harms
Identification of disease predisposition or early diagnosis, leading to prevention or effective therapy.	If no treatment or prevention available, diagnosis may cause anxiety/depression. False-positives leading to more testing; some testing may be invasive or have side effects (biopsies, surgeries, anxiety, depression). Incidental findings/indolent disease ^1^ (over-diagnosis, over-treatment, and some treatments may be invasive, have serious side effects, and be costly). Harms of testing (e.g., radiation, bleeding, colon perforation). Cost-effectiveness.

^1^ Incidental finding is defined as a finding that is unrelated to the primary reason of patient testing. Indolent disease is defined here as a disease detected by screening that would have otherwise not been detected in a patient’s lifetime. Reprinted with permission from ref. [40]; published by De Gruyter, 2016.

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
