# Peer review of "Can Circulating Tumor DNA Support a Successful Screening Test for Early Cancer Detection? The Grail Paradigm"

_diagnostics, 2021, doi:10.3390/diagnostics11122171_

Round 1

Reviewer 1 Report

This is an important commentary on a highly relevant topic. 

I have 2 min or suggestions:

  1. The Reference "c" in table 1, column 7 seems not correct
  2. The authors should give exact calculations on PPV and number needed to screen and false positives deduced from known numbers on prevalence and the published sensitivity/specificity of the Grail  and Cancer Seek techniques so that readers have an impression about the numbers in a real-world screening scenario.

Author Response

Response to Reviewer 1 Comments

Point 1: The Reference "c" in table 1, column 7 seems not correct.

Response 1: Corrected, as suggested

Point 2: The authors should give exact calculations on PPV and number needed to screen and false positives deduced from known numbers on prevalence and the published sensitivity/specificity of the Grail  and Cancer Seek techniques so that readers have an impression about the numbers in a real-world screening scenario.

Response 2: We do provide PPV for Grail under a screening scenario, at fixed sensitivity of 10%, fixed specificity of 99.5% and two prevalences (0.1% and 1%). Since the grail test targets 50 tumor types, calculating PPV or every cancer would be impractical. Our statement below captures our main conclusion

If the sensitivity is assumed to be 10% for early cancers under a screening scenario, at 99.5% specificity, and a hypothetical prevalence of cancer in the general screening population of either 1% (for fairly common cancers) or 0.1% (for fairly rare cancers), the PPV will be 17% in the first case and 1.7 % in the second case. We seriously question that a successful screening program for cancer can be sustained with such low PPVs. The false positivity rates will lead a rather large number of non-cancer patients who will undergo additional, unnecessary and probably harmful testing [40].

Reviewer 2 Report

This nice and interesting review shows the limitations of liquid biopsy in particular as a screening tool to detect small tumors. Overall the manuscript is well written and well documented. I have two minor comments:

  • Authors should provide some details on how DNA is released and accordingly whether we expect differences between tumor types in terms of quantity of ctDNA with some tumors releasing fewer or no DNA compared to others
  • I fully agree with the limtations of liquid biopsy in detecting small tumors. In fact do we need such tool as we might be in a situation where the liquid biopsy detects ctDNA while no evidence is found on radiologic investigations? This would be particularly useless as treatment of small tumors relies mostly on surgery which could obviously not been performed without radiologic findings. Authors should also elaborate on this point.

Author Response

Response to Reviewer 1 Comments

Point 1: Authors should provide some details on how DNA is released and accordingly whether we expect differences between tumor types in terms of quantity of ctDNA with some tumors releasing fewer or no DNA compared to others

Response 1: The point raised by the reviewer is addressed in our cited reference 7. The same issue is also discussed briefly in Ref [34]. Although the issue has theoretical importance, in the context of diagnostics is not a decisive factor2. 

Point 2: I fully agree with the limtations of liquid biopsy in detecting small tumors. In fact do we need such tool as we might be in a situation where the liquid biopsy detects ctDNA while no evidence is found on radiologic investigations? This would be particularly useless as treatment of small tumors relies mostly on surgery which could obviously not been performed without radiologic findings. Authors should also elaborate on this point.

Response 2: A very important issue. We added an explanatory note as suggested.